# Ultrasound-Assisted Encapsulation of Citronella Oil in Alginate/Carrageenan Beads: Characterization and Kinetic Models

**Aji Prasetyaningrum \*, Bangkit Suryo Wicaksono, Azafilmi Hakiim, Aulia Dwi Ashianti, Sadrakh Farel Christian Manalu, Nur Rokhati, Dani Puji Utomo** 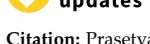 **and Mohammad Djaeni**

Department of Chemical Engineering, Faculty of Engineering, University of Diponegoro, Semarang 50275, Indonesia
* Correspondence: aji.prasetyaningrum@che.undip.ac.id

**Abstract:** The objective of this research was to investigate the effect of ultrasonication on citronella oil encapsulation using alginate/carrageenan (Alg/Carr) in the presence of sodium dodecyl sulfate (SDS). The functional groups of microparticles were characterized using Fourier transform infrared spectroscopy (FTIR), and the beads' morphologies were observed using a scanning electron microscope (SEM). The FTIR results showed that the ultrasonication process caused the C-H bonds ($1426 \text{ cm}^{-1}$) to break down, resulting in polymer degradation. The SEM results showed that the ultrasonication caused the presence of cavities or pores in the cracked wall and a decrease in the beads' size. In this study, the use of ultrasound during the encapsulation of citronella oil in Alg/Carr enhanced the encapsulation efficiency up to 95–97%. The kinetic evaluation of the oil release of the beads treated with ultrasound (UTS) showed a higher $k_1$ value of the Ritger–Peppas model than that without ultrasonication (non-UTS), indicating that the oil release rate from the beads was faster. The R/F value from the Peppas–Sahlin model of the beads treated with UTS was smaller than that of the non-UTS model, revealing that the release of bioactive compounds from the UTS-treated beads was diffusion-controlled rather than due to a relaxation mechanism. This study suggests the potential utilization of UTS for controlling the bioactive compound release rate.

**Keywords:** encapsulation; ultrasound; citronella oil; alginate; carrageenan



## 1. Introduction

*Citronella (Cymbopogon nardus)* belongs to the Panicodiae family of Graminales and is widely used in pharmaceuticals, the food/beverage industries, fragrances, and as cosmetic ingredients [1–3]. Citronella has pharmacological activities, namely anti-fungal, anti-bacterial, anti-inflammatory, antioxidant, and anti-virus activities [4–6]. *Citronella* oil is yellowish and mainly contains citronellal, geraniol, and citronellol [7]. However, the bioactive compounds of citronella also have unstable properties such as volatility, rapid release, low solubility, and are easily damaged in cases of environmental stress, as well as under conditions related to processes such as pH, temperature, oxygen, exposure to light, and time storage [8,9]. The encapsulation of bioactive compounds could be an alternative way to preserve the bioactivity of citronella oil. Encapsulation is the process of entrapping a compound in an outer layer or matrix. This process serves to increase the physical activity, solubility, and bioavailability of bioactive compounds and is a well-accepted method for enhancing their efficacy [10].

Alginate is a kind of natural biocompatible and biodegradable polymer material and has been widely used in pharmaceuticals, the food industry, tissue engineering, etc. [2,11]. Alginates (Alg) have the ability to form hydrogels with the addition of divalent and polyvalent metal ions, especially in the presence of calcium ions [12]. Alginates have been extensively studied for drug delivery applications because they have a high encapsulation

efficiency and do not form agglomerates in organs during drug delivery [13]. However, alginate has a drawback in that the gel matrix is porous, highly permeable, and easily degraded, so it is difficult to control the release of an encapsulated material from it [14]. Therefore, a combination of different biopolymers is the best option to overcome the weaknesses of alginate.

Biopolymers that are often used in encapsulation include alginate and chitosan. In the formation of micro-polymers, alginate-based formulations are used as texture modifiers, stability enhancers, and enhancers of the long-term efficacy of active compound due their biostability, biocompatibility, biodegradable nature, and non-toxic delivery system. However, the mixture of alginate and chitosan is not soluble at pH 7, but it is soluble at pH 5 [14]. This means that one of the potential polymers that can be combined with alginate at pH 7 (neutral) is carrageenan (Carr). The addition of carrageenan to alginate has been reported to increase the porosity of the hydrogel [13]. Moreover, it also increases the stability of the hydrogel formation and its encapsulation efficiency as well as its swelling value. According to Ramdhan et al. [14], the addition of kappa carrageenan into alginate can produce a soft/elastic hydrogel that is dependent on the concentration of kappa carrageenan (Carr). Yu et al. [15] reported that the incorporation of alginate/carrageenan hydrogels into $CaCl_2$ produces a network structure with a high affinity. In addition, in the presence of electrostatic attraction, alginate and $Ca^{2+}$ form a bond network. At the same time, adjacent $Ca^{2+}$ and carrageenan ($OSO^{3-}$) form crosslinking networks between macromolecular crosslinking networks. Obvious results were also shown in a previous study by Postolovic et al. [12], who performed the encapsulation of curcumin into a combination of alginate and carrageenan, where the study resulted in a high encapsulation efficiency of up to 95.74%.

Encapsulation using conventional methods is usually obtained by first preparing emulsions of bioactive components with a wall-constructing material. The formation of the core–skin material in the conventional method is generally more complex. Additional materials such as chemical crosslinkers and emulsifiers are required in most cases. To produce hydrogels, calcium chloride ($CaCl_2$) is commonly used as a crosslinking agent to induce gelatinization due to its ability to form intermolecular bonds between alginic acid and calcium ions. Szekalska et al. [16] proved that a gel formation cross-linked with $Ca^{2+}$ can improve polymer stability.

The addition of surfactants into proto-bead solutions are required to increase the stability of the wall material. In the encapsulation process, the surfactants that are often used include Tween 80, Poloxamer 188, Span 20, and SDS (sodium dodecyl sulfate) [17–19]. The most commonly used sulfate surfactant is SDS, which is the most-studied anionic surfactant known to science. SDS is an organic compound that is commercially found in powder or pellet form, is soluble in water, and has a low toxicity. The surfactant SDS has an amphiphilic molecule containing both hydrophilic and hydrophobic moieties. Previous research has shown that SDS is not carcinogenic when applied directly to the skin or consumed [20]. In addition, SDS is also able to increase the thickness of films and increase the encapsulation efficiency [19,21].

Furthermore, core–skin materials synthesized by conventional methods have been reported to have low stability, and their droplet size is larger and inhomogeneous [22]. To overcome the problems of the conventional method, ultrasonic treatment can be used. The use of ultrasound (UTS) can effectively reduce the droplet size of the emulsion to the nano-size level [23]. Applications of ultrasound technology utilize high-frequency sonic waves of more than 20 kHz to generate agitation and are useful in the formation of microcapsules [24,25]. Agitation creates gas or liquid cavities within the system that are capable of exploding. The collapse of the cavitation bubble causes the $H_2O$ bond to break and generate free radicals. These radicals can then trigger the ability for microcapsule formation [22]. In addition, the use of ultrasound can reduce the homogenization time compared to conventional methods. The decrease in the droplet size of the emulsion can improve the encapsulation efficiency of the bioactive compounds in the wall material due

to the nanosized microcapsule size [24]. Sangolkar et al. [17] studied the encapsulation of peppermint flavor in gum arabic, which was assisted by ultrasound (UTS) and resulted in a smaller droplet size and a better emulsion. Liu et al. [26] also conducted ultrasound-assisted resveratrol encapsulation in zein–gum arabic, which resulted in smaller and more uniform particle sizes, leading to a high encapsulation efficiency.

Emulsification and depolymerization can be promoted by UTS exposure to form polymer cross-linkages, resulting in the formation of polymer microspheres [27]. Among the many applications, emulsion formulation has been successfully carried out using UTS, which results in low polydispersity (homogeneous), size reduction, high accuracy, and no particle aggregation [28]. In a sonochemical approach, this method offers great potential and versatility, where the emulsification and crosslinking processes are achieved in one step. Experiment using the sonochemical technique are relatively simple, and stable core–shell materials can be produced in a short time. In addition, there is a greater flexibility in choosing different core and shell materials when using this technique [23].

Since studies related to the encapsulation of citronella oil using alginate/carrageenan (Alg/Carr) by UTS treatment are scanty, the objective of this study was to investigate the characteristics of citronella-oil-encapsulated beads obtained from UTS-treated biopolymers (Alg/Carr) in terms of their stability and bioavailability. These polysaccharides were ionically cross-linked with $CaCl_2$. Furthermore, the effect of the ultrasonication on the encapsulation efficiency was also evaluated. The morphological properties of produced beads and their functional groups were characterized by SEM and FTIR, respectively. The release kinetics behavior of bioactive compounds was also studied for further studies of controlled release. Three different mathematical models were proposed to describe the release mechanism. The physical stability of the citronella-oil-loaded capsules was also examined.

## 2. Experimental

### 2.1. Materials

Citronella oil was obtained from Intan Chemical Ltd., Surabaya, Indonesia, with the following specifications: relative density at 20/20 °C: 0.87–0.89, and carbonyl compounds content > 75%. Chemicals such as sodium alginate (molar mass 216.12 g/mol, molar weight 90–180 kDa with CAS Number 9005-38-3SIGMA-Aldrich, St. Louis, MO, USA), carrageenan (molar weight 193–324 kDa), calcium chloride ($CaCl_2$), sodium dodecyl sulfate (SDS), hydrochloric acid (HCl), sodium hydroxide (NaOH), and buffer solution (Merck Chemical Co., Darmstadt, Hesse, Germany) were obtained and were of analytical grade.

### 2.2. Preparation of the Alg-Carr Biopolymeric Capsules

Citronella-oil-loaded Alg/Carr nanoemulsion was prepared according to methods reported by Elgegren et al. [29] and Iurciuc-Tincu et al. [30] with some modifications. The encapsulation procedure is illustrated in Figure 1. The particles were prepared in two steps: In the first step, the optimal Alg/Carr ratio used in this study was determined to be 2:1 [12]. The alginate solutions were prepared by dissolving 2% (*w/v*) sodium alginate using pure water and then homogenizing the solution with a magnetic stirrer for 30 min at ±27 °C. Then, 0.7% (*w/v*) SDS surfactant and 16% citronella oil (*v/v*) were added into the alginate solution. The solution was continuously mixed using a magnetic stirrer for 20 min. Then, the mixed solution was sonicated using an ultrasonic transducer. The following different ultrasonication times were examined: 0 min, 4 min, 6 min, 8 min, 10 min, and 12 min.

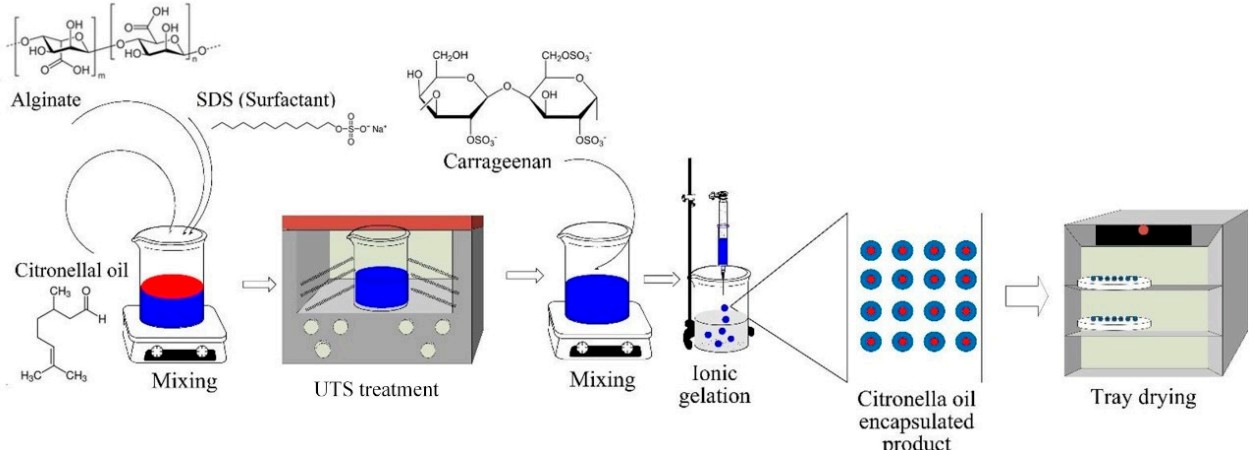

**Figure 1.** Experimental setup of ultrasound-assisted encapsulation.

In the second step, 1% (*w/v*) carrageenan was dissolved in distilled water and homogenized at 28 °C with a magnetic stirrer for 30 min. The solution obtained from the first step was added into the carrageenan solution with a volume ratio of 1:1 (*v/v*). The mixture was then mixed using a magnetic stirrer for 15 min until homogonous solution was obtained. For beads generation, the prepared biopolymeric solution was dripped into a 0.2 M $CaCl_2$ solution using a syringe. The beads were immediately formed in $CaCl_2$ solution and left for 30 min to obtain the constituent stable beads. The beads were then separated from the $CaCl_2$ solution and dried in a tray dryer at 27 °C for 48 h.

### 2.3. Characterization of the Particles

### 2.3.1. Scanning Electron Microscope (SEM) Analysis

The prepared Alg/Carr capsule beads with and without ultrasonication were analyzed for their surface morphology using a SEM JEOL JSM-6510LA device. Prior to analysis, the samples were metalized with gold using a sputter coating unit. The SEM magnification was 7500×.

### 2.3.2. Fourier Transform Infrared Spectroscopy (FTIR) Analysis

The results of the Alg/Carr encapsulation after ultrasonication were analyzed with a Perkin Elmer Spectrum IR 10.6.1 spectrophotometer (Perkin Elmer Inc., Waltham, MA, USA) to determine the functional groups in the 4000–450 $cm^{-1}$ spectrum. The infrared (IR) spectra were obtained by mixing the dried samples (~1 mg) with dried KBr (100 mg).

### 2.3.3. Encapsulation Efficiency

Encapsulation efficiency was determined by measuring the amount of untrapped bioactive compounds in the Alg/Carr beads after being dripped into the $CaCl_2$ solution. This measurement was evaluated by an indirect method. A total of 2 mL of $CaCl_2$ solution was taken, and then 2 mL of $AlCl_3$ and 2 mL of $KCH_3COO$ were added. After that, the bioactive compounds content was determined using a Genesys 20 UV-vis spectrophotometer at a wavelength of 367 nm. In this case, the encapsulation efficiency was determined as per Cirri et al. [31]:

$$EE\ (\%) = \frac{Qt\ -\ Qr}{Qt} \times 100 \qquad (1)$$

where Qt is the amount of bioactive compounds in the citronella oil and Qr is the bioactive citronella oil present in the $CaCl_2$ solution after encapsulation.

2.3.4. Bioactive Compounds Release Kinetics Study

The release of citronella oil in the encapsulation was carried out at a pH of 1.2 and 6.8. A total of 0.2 g of encapsulated citronella oil in the best formulation was put into 30 mL of a solution with pH 1.2 and 6.8 and soaked for 24 h. For each time interval, 5 mL of sample was taken to be analyzed by a spectrophotometer at a wavelength of 367 nm to determine the release of the bioactive compounds in the citronella oil at each time. Three different mathematical models were used to determine the release mechanism of the citronella oil in the beads. In calculating all models, experimental data was needed from the amount of bioactive compounds released at equilibrium time ($M_{eq}$) and time t ($M_t$), which was then calculated using the empirical equation below:

**The Higuchi equation**

$$\frac{M_t}{M_{eq}} = k_h t^{0.5} \tag{2}$$

where $k_h$ is the Higuchi coefficient. If the correlation coefficient value is higher for the fitting, it can be interpreted that the main mechanism of the release of bioactive compounds is a diffusion release mechanism [32,33].

**Ritger–Peppas model**

$$\frac{M_t}{M_{eq}} = k_1 t^n \tag{3}$$

where the value of $k_1$ is the release rate constant based on the structural and geometric characteristics of the release system, and *n* is the diffusion exponent. If $n \leq 0.43$, the release mechanism is a molecular-diffusion-based bioactive compound release mechanism, whereas, if $n \geq 0.85$, it is a relaxation-based bioactive compound release mechanism during the bioactive transport mechanism, which includes tension and state transitions in the polymer swells in water. If $0.43 < n < 0.85$, then the release of the bioactive compounds is controlled by diffusion, followed by relaxation [32].

**Peppas–Sahlin model**

$$\frac{M_t}{M_{eq}} = k_1 t^m + k_2 t^{2m} \tag{4}$$

This release kinetics model is applied to the diffusion and relaxation release mechanism in drug release processes. To determine the release of a bioactive compound, the F value represents the contribution of Fickian diffusion, while the R value represents the relaxation contribution. The R/F ratio indicates the contribution of relaxation and Fickian diffusion to the drug release. When R/F = 1, the release mechanism is contributed to equally by erosion (relaxation) and diffusion. If R/F > 1, relaxation (erosion) dominates, while for R/F < 1, diffusion dominates. The relaxation ratio (R) vs. the Fickian contribution (F) can be calculated as follows:

$$\frac{R}{F} = \frac{k_2}{k_1} t^m \tag{5}$$

where $k_2$ is the relaxation kinetic constant obtained from the Peppas–Sahlin equation, and $k_1$ is the Fickian diffusion rate constant.

**3. Result and Discussion**

*3.1. Citronella Oil Composition Analysis Using GC–MS*

The chromatogram of the citronella oil components identified using GC–MS is shown in Figure 2. The data in Table 1 presents the identified compounds of the citronella oil components. The GC–MS results showed that there were 15 components contained in the citronella essential oil. There were two main compounds of the citronella oil contained in the sample, namely citronella at 19.77% with a retention time of 16.30 min and geraniol at 17.98% with a retention time of 29.92 min. In addition, other compounds had a high % area, including α-pinene at 19.57% with a retention time of 4.06 min and neryl acetate at 20.60% with a retention time of 26.95 min. Other minor components were identified at percentages below 10%, such as camphene, delta 3-carene, 1-methyl-4-(1-methyleneallyl)cyclohexene,

linalool, β-elemene, trans-caryophyllene, citronellil acetate, z-citral, α-amorphene, and elemol. The percentage was acceptable, as it was similar to that reported by Rastuti et al. [34] reporting the component of citronella (36.63%) and geraniol (25.71%). Citronella and geraniol compounds tend to be semipolar compounds. During their separation, the interaction of the two compounds with the GC column (nonpolar) was only affected by the boiling point of the compounds. Citronella and geraniol had retention times that tended to be short because the two compounds have the same polarity and interacted weakly with the GC column. Citronella has a lower boiling point than geraniol because the -OH group on geraniol can form strong hydrogen bonds, so its boiling point is higher. Citronella is a type of monoterpenoid aldehyde (-CHO). Therefore, citronella had a faster retention time than geraniol [35].

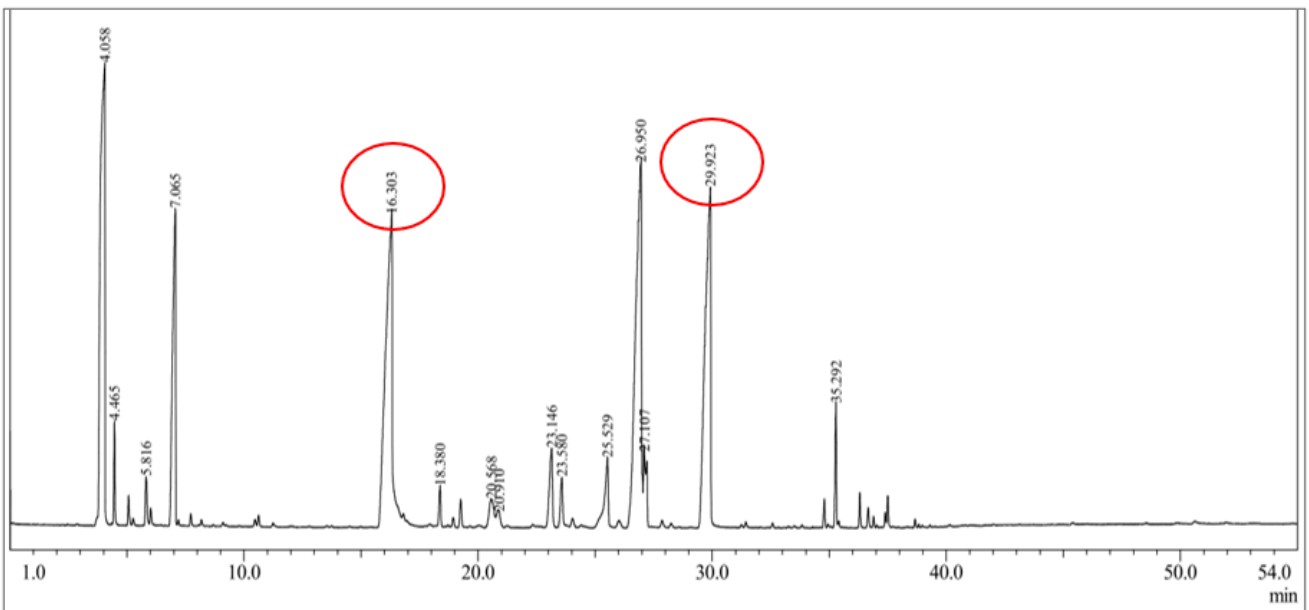

**Figure 2.** GC–MS chromatogram of citronella oil.

**Table 1.** List of identified chemical compounds in citronella oil.

| Peak# | R.Time (min) | Percentage (%) | Name |
|-------|--------------|----------------|------|
| 1 | 4.06 | 19.57 | α-pinene, (-)- |
| 2 | 4.46 | 0.95 | Camphene |
| 3 | 5.82 | 0.74 | Delta 3-carene |
| 4 | 7.06 | 8.72 | 1-Methyl-4-(1-methyleneallyl)cyclohexene |
| 5 | 16.30 | 19.77 | Citronella |
| 6 | 18.38 | 0.64 | Linalool |
| 7 | 20.57 | 1.34 | β-elemene |
| 8 | 20.91 | 0.62 | Trans-caryophyllene |
| 9 | 23.15 | 2.32 | Citronellyl acetate |
| 10 | 23.58 | 1.05 | Z-citral |
| 11 | 25.53 | 1.81 | Z-citral |
| 12 | 26.95 | 20.60 | Neryl acetate |
| 13 | 27.11 | 2.13 | α-amorphene |
| 14 | 29.92 | 17.98 | Geraniol |
| 15 | 35.29 | 1.77 | Elemol |

### 3.2. Characterization of Citronella Oil Beads by FTIR

The functional properties of the citronella oil microcapsules were investigated using FTIR. The spectra of the citronella-oil-containing microcapsules are presented in Table 2. The FTIR spectra of the UTS-treated microcapsule beads were compared with conventional microcapsule beads (with and without surfactant addition), empty Alg/Carr beads, and citronella oil for a comprehensive and in-depth understanding.

**Table 2.** FTIR spectra.

| Wavenumber Range (cm$^{-1}$) | Functional Groups | Reference |
|:---:|:---:|:---:|
| 2935–2915 | C-H | [36] |
| 2865–2845 | C-H | [36] |
| 1690–1800 | C=O | [37] |
| 1499≈ | C-O-S | [20] |
| 1370–1485 | C-H | [36] |
| 1210–1260 | S=O | [38] |
| 895–885 | C-H | [36] |

The FTIR spectra of the citronella oil from Figure 3a exhibited IR absorptions at 1730 cm$^{-1}$, 2368 cm$^{-1}$, and 2922 cm$^{-1}$, which were associated with symmetrical and asymmetrical C=O vibrations and C-H strains of the methyl groups of the aromatic compounds in the citronella oil. The absorption band at 2368 cm$^{-1}$ provided evidence indicating the presence of an aldehyde group. In the spectrum of the Alg/Carr from Figure 3b, the absorption at 891 cm$^{-1}$ was assigned to the anomeric C-H groups of β-galactopyranose. The peak spectrum at 1241 cm$^{-1}$ was due to the stretching of the sulfate group S=O, which belonged to the Carr. There was also an intense band at 1094 cm$^{-1}$ caused by the valence vibration of the C–O bond. An absorption band was also seen at 1527 cm$^{-1}$, which was due to the asymmetric and symmetrical vibrations of the carboxylate anion of the Alg. The peak of the 1020 cm$^{-1}$ wave was due to the C-O absorption band, which is a group owned by the Alg.

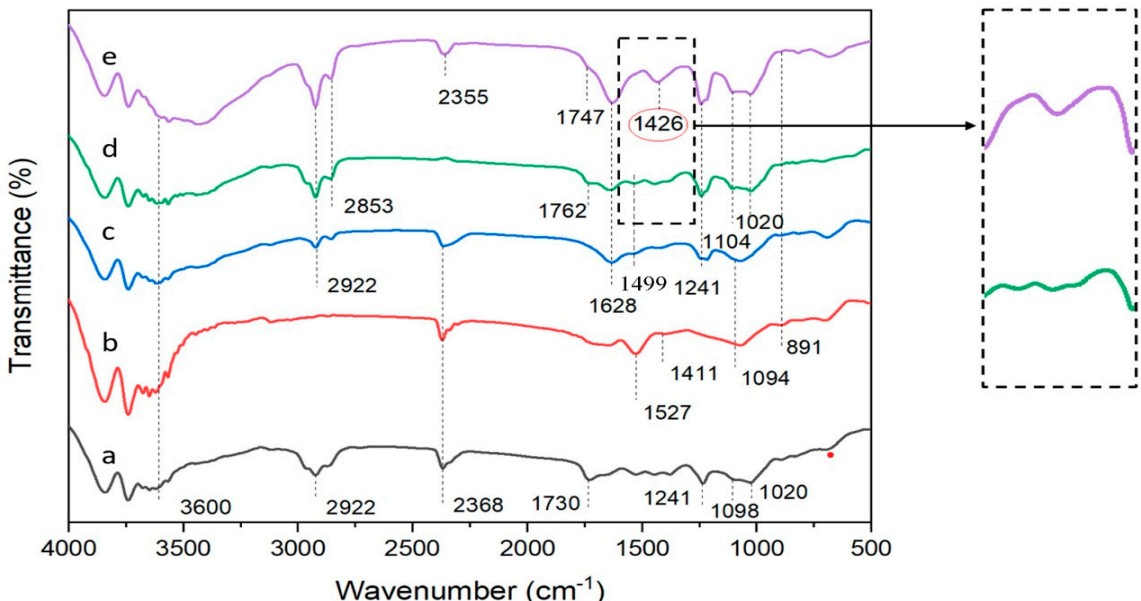

**Figure 3.** FTIR spectra for: (**a**) citronella oil; (**b**) Alg/Carr; (**c**) Alg/Carr/SDS; (**d**) Alg/Carr/SDS/citronella oil (non-UTS); and (**e**) Alg/Carr/SDS/citronella oil (UTS).

As shown in Figure 3c, with the addition of the SDS into the Alg/Carr, a peak spectrum was formed at 1499 cm$^{-1}$, which was due to a C-O-S sulfate group belonging to SDS containing a spectrum of beads. As shown in Figure 3d,e, it was determined that the adsorption band widths between 2853 cm$^{-1}$, 1762 cm$^{-1}$, and 1747 cm$^{-1}$ were due to the vibrations of the C=C and C-H bonds stretching the methyl groups of the aromatic compounds in the citronella oil, and this proved that citronella oil smells good and was successfully packaged in the Alg/Carr wall material. As shown in Figure 3d,e, the properties of the functional groups of the citronella oil were found to have been little effected by the ultrasound treatment. As shown in the spectrum from Figure 3e, the Alg/Carr/SDS/citronella oil treated with UTS showed a sharp band at 1426 cm$^{-1}$, which showed the C-H bond. The ultrasonication process caused the C-H bond to break, resulting in polymer degradation. Furthermore, the citronella oil was encapsulated in the Alg/Carr wall materials in the presence of the ultrasonication. Therefore, the ultrasonication had no significant effect on the FTIR spectrum encapsulation of the citronella oil in this process. This was in accordance with the research conducted by Liang et al. [39], where stronger hydrogen bonds and electrostatic interactions caused changes in the secondary structure of zein in resveratrol encapsulation.

### 3.3. Morphological Properties of Citronella Oil Beads by SEM

SEM images were taken to investigate the morphologies of the prepared microcapsule beads. Figure 4 shows the surface morphology of the alginate/carrageenan beads investigated by SEM at a magnification of 7500×.

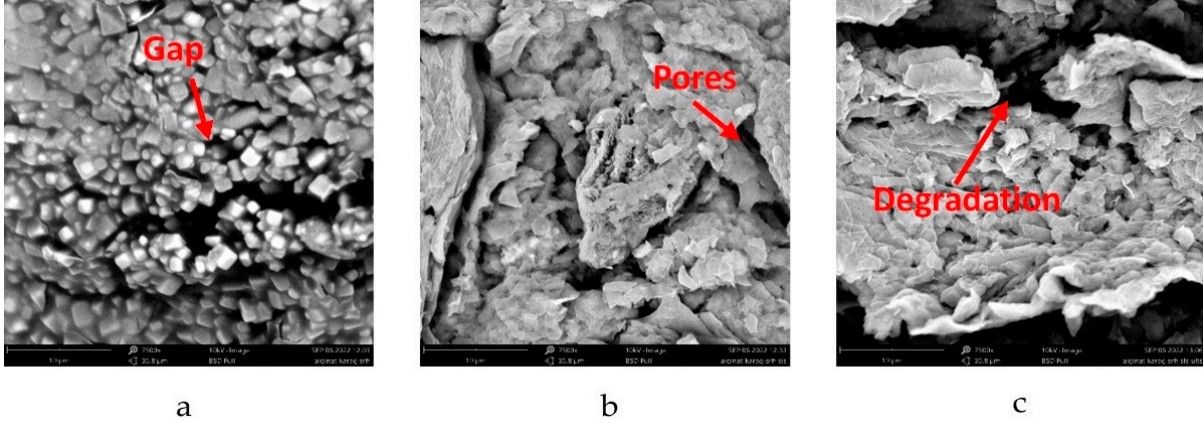

a         b         c

**Figure 4.** SEM images of citronella-oil-encapsulated beads: (**a**) without SDS/non-UTS; (**b**) SDS/non-UTS; and (**c**) SDS/UTS.

According to Figure 4, the Alg/Carr encapsulating materials were able to trap the citronella oil. Alg/Carr are hydrophilic compounds, while citronella oil is a hydrophobic compound, which meant that the emulsion formed could not be mixed properly. This caused gaps in the particles, as shown in Figure 4a. The presence of gaps in the particles caused the trapping efficiency to be poor, so an emulsifier was needed. In this study, SDS surfactant was added as an emulsifier. The SEM results showed that the addition of SDS significantly changed the morphology of the irregular beads and the presence of cavities or pores, which can be seen in Figure 4b. This phenomenon was similar to the study conducted by Kaygusuz et al. [21], where the beads showed that their surface turned into a leafy structure with many cavities. According to this model, surfactants were bound to the polymer chains by forming aggregates to form necklace-like complexes.

Figure 4c (with ultrasound treatment) shows a different morphology compared to that obtained without ultrasound (Figure 4b). Significant damage to the grain surface was detected for the sonicated samples, which could be observed as dark pores and elongated fissures. Cracks and pores on the surface of the sonicated beads were formed by

cavitation phenomena, which locally induced very high stresses and shear forces, leading to the mechanical degradation of the amorphous layer wall material, thereby reducing the particle size, which mainly resulted in multiple channels, fissures, and cracks on the grain surface. This further broke the glycosidic bonds and degraded the carboxyl groups of the alginate, thereby loosening the tight structure of the particles, which in turn provided more space for the surrounding medium to penetrate the particles. Similar results were also shown in a study conducted by Falsafi et al. and Wardhani et al. [40,41].

The ultrasonication-induced depolymerization may have been complicated by the fact that the resulting polymer could also be simultaneously broken down by the strong shear forces resulting from the collapsing bubbles, thereby causing the polymer bonds to break and lowering the emulsion viscosity. The decrease in the viscosity caused the particle size to get smaller with an increase in surface area. This can be seen in Figure 5, where the longer the ultrasonication time, the smaller the particle size. This phenomenon was reflected in the FTIR absorption spectrum in the form of a sharper peak at 1426 cm$^{-1}$ and was in accordance with the statement of Liu et al. [26].

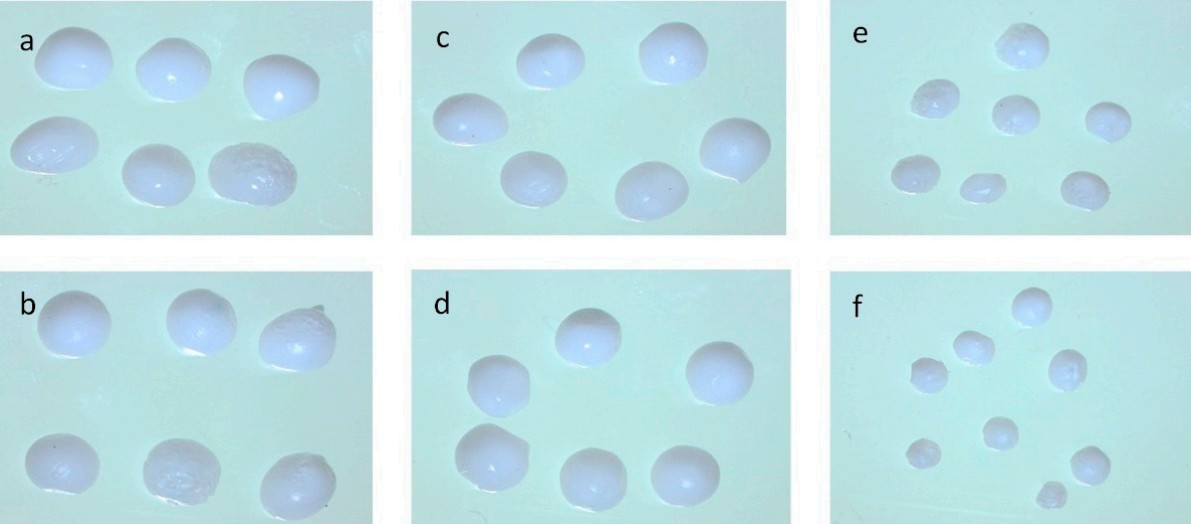

**Figure 5.** Optical image of citronella-oil-encapsulated beads with varying time: (**a**) 0 min; (**b**) 4 min; (**c**) 6 min; (**d**) 8 min; (**e**) 10 min; and (**f**) 12 min.

### 3.4. Encapsulation Efficiency and Particle Size

The amount of bioactive compounds in the citronella oil trapped in the Alg/Carr beads depended on the encapsulation efficiency. Table 1 shows that the UTS time significantly affected the particle size. The encapsulation without ultrasound (0 min) obtained a particle size of 3.27 mm, while the beads obtained using UTS for 12 min obtained a particle size of 1.56 mm. UTS can reduce intermolecular steric repulsion and resistance, create a more compact particle structure, and reduce particle sizes [26]. A similar phenomenon of size reduction due to UTS time was reported in the study of Zaghian and Goli [42]. Simultaneously, the shear force generated by the bursting of bubbles caused by ultrasonic cavitation can break the large polymers formed, thereby effectively reducing the molecular weight of the final particles, making the molecular weight distribution of the particles narrower, and ultimately lowering the particle size [27]. The electrostatic charge of the polymer also played an important role in the encapsulation formation and the citronellal oil charging and release processes. Alginate and carrageenan have strong negative charges due to the presence of free carboxyl and hydroxyl groups as well as the sulfated galactan in the carrageenan. The negative sites of the Alg/Carr interacted with Ca$^{2+}$ to form a polymer network in the formation of the encapsulation. The charging of citronellal oil into the biopolymer encapsulation was assisted by the presence of the surfactant due to its amphiphilic properties. The hydrophilic moiety of the surfactant interacted with the ionic

sites of the Alg/Carr; on the other hand, the hydrophobic part of the surfactant attracted the citronellal oil molecule. These hydrophilic/hydrophobic properties also controlled the release of the citronellal oil from the Alg/Carr matrix [43]. Furthermore, the efficiency of the encapsulation was influenced by the irradiation from the UTS.

The encapsulation efficiency increased by looking at the exact time from 0 to 10 min. At minute 10, the highest encapsulation efficiency reached 97.74%. The encapsulation efficiency increased with the UTS treatment. Without the UTS treatment, the efficiency obtained was 95.82%; with the UTS treatment, the encapsulation efficiency was able to reach 97.74%. From 10 to 12 min, the encapsulation efficiency decreased as the UTS time increased. This could have happened because the sonication process was able to open the polysaccharide structure, and more glycosidic bonds were broken. Excessive UTS treatment can lead to the unfolding and aggregation of polysaccharides, resulting in a decreased encapsulation efficiency [44]. The sonication caused a decrease in the particle size and an increase in the surface area, which made the coating process easier and provided a high encapsulation efficiency [45]. On the other hand, increasing the sonication time to 12 min led to a decrease in the encapsulation efficiency (97.55%), indicating that excessive sonication could lead to polymer degradation leading to a decrease in its ability to completely coat the particles. A longer UTS time led to shorter polymer chains and lower particle sizes. This affected the strength of the intermolecular bonds and reduced the ability of the Alg/Carr to hold the bioactive citronella oil within the beads and release the bioactive compounds. A similar phenomenon of encapsulation efficiency vs. UTS time was reported in the study of Luo et al. [46] and Wardhani et al. [41]. A prolonged application of ultrasonication treatment is not recommended, as it may damage the bioactive compounds present in the formulation.

UTS can provide a stabilizing effect in oil-in-water or water-in-oil emulsions by increasing the stability of the emulsion. The efficiency and stability index of the ultrasound-assisted encapsulation emulsion depends on the effectiveness of the cavitation mechanism. Generally, high-intensity UTS is easy to operate and requires only a small amount of surfactant. To promote a stable emulsion, the generator generates electrical energy with a frequency generally in the 20–100 kHz range. One of the most common problems in ultrasound-assisted emulsion formation is overprocessing. This phenomenon can occur when the homogenization process has a higher energy density than the physicochemical limit imposed by the emulsifier and results in an increase in droplet size due to recalescence [47].

*3.5. Release Kinetics of Citronella Beads*

Figure 6 shows a graph of the percentage of the cumulative release of the citronella oil from the Alg/Carr carried out in simulated gastric (pH 1.2) and intestinal (pH 6.8) fluids for 24 h. From the experimental results, it was shown that the %release of the citronella oil at pH 1.2 (Figure 6a,b) was less than that at pH 6.8 (Figure 6c,d). This was also shown in the results presented in Table 3, where the value of k (release rate) at pH 1.2 was smaller than at pH 6.8. This phenomenon occurred as a result of the alginate being more soluble at a higher pH due to the formation of carboxyl ion groups. Alginate is insoluble and dissociates under acidic conditions, indicating that alginate had the potential to protect the content of citronella oil in the gastric environment.

The results of the release kinetics study showed that the encapsulated beads using UTS (Figure 6b,d) exhibited a faster release ability than the non-UTS-treated beads (Figure 6a,c). This was because the UTS treatment broke the polymer bonds, so that the ability of the polymer bonds to protect the citronella oil decreased. This phenomenon also occurred in a study conducted by Wardhani et al. [41].

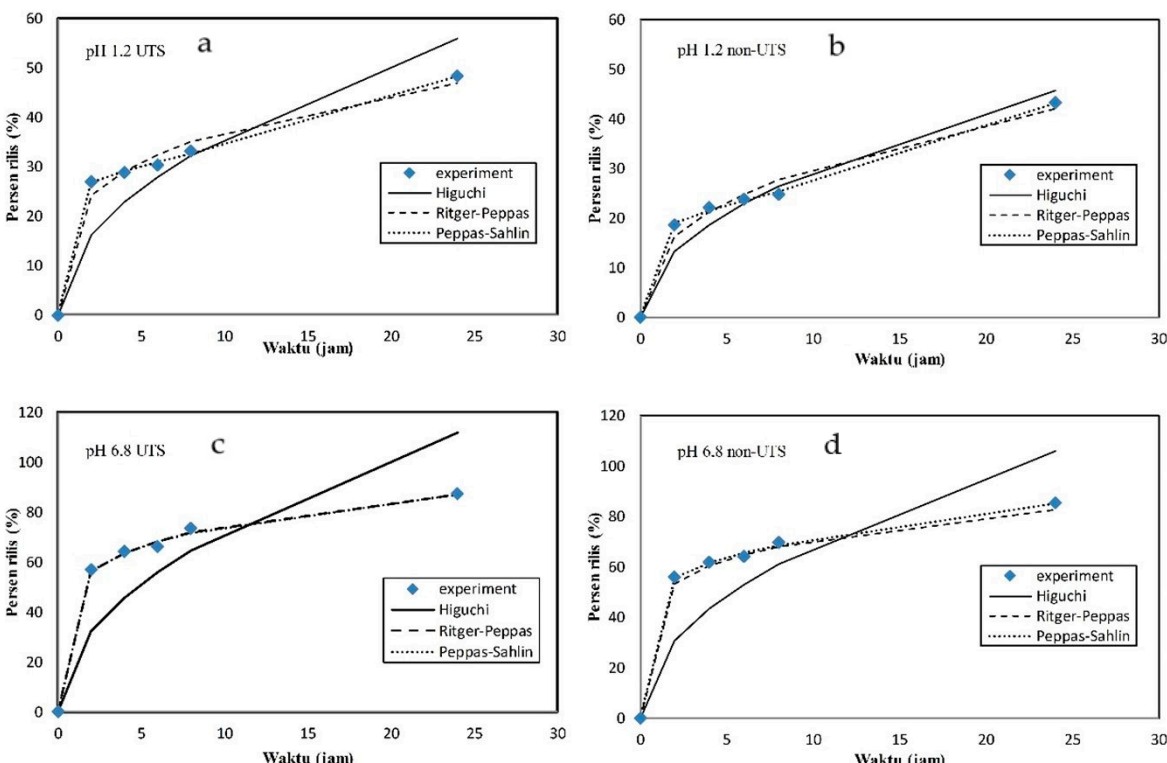

**Figure 6.** Release kinetic plots: (**a**) pH 1.2 non-UTS; (**b**) pH 1.2 UTS; (**c**) pH 6.8 non-UTS; (**d**) pH 6.8 UTS.

**Table 3.** Encapsulation efficiency and particle size.

| Ultrasound Time (min) | Diameter (mm) | Encapsulation Efficiency (%) |
| --- | --- | --- |
| 0 | 3.27 ± 0.13 | 95.82 ± 0.13 |
| 4 | 3.15 ± 0.03 | 97.17 ± 0.21 |
| 6 | 2.88 ± 0.06 | 97.42 ± 0.10 |
| 8 | 2.81 ± 0.03 | 97.48 ± 0.14 |
| 10 | 2.10 ± 0.08 | 97.74 ± 0.16 |
| 12 | 1.56 ± 0.04 | 97.55 ± 0.10 |

The kinetic parameters obtained from the hydrogel beads in the alginate/carrageenan UTS and non-UTS samples were obtained after the occurrence of data with three mathematical model equations: Ritger–Peppas, Peppas–Sahlin, and Higuchi. It can be seen from Table 4 that the compounds in the citronella oil were released from all the fitting hydrogel beads from all the models, and the statistically best for the media were the obtained using the Ritger–Peppas model and the Peppas–Sahlin model. The best statistical indication of fitting was observed with large correlation coefficient values obtained in the Ritger–Peppas model and the Peppas–Sahlin model. The Higuchi model showed a lower value than the Ritger–Peppas model and the Peppas–Sahlin model, and this was because most of the release mechanisms of the bioactive compounds were caused by diffusion. The fitting value of the Higuchi model was not as good as that of the Ritger–Peppas model and the Peppas–Sahlin model, which means that the relaxation factor could not be completely removed from the mechanism, followed by removing the citronella oil compound from the Alg/Carr beads. A similar assumption could be obtained with the best-fitting result of the Ritger–Peppas model, where a case value of $n \leq 0.43$ in all the hydrogel beads in the media indicated Fickian diffusion to be the dominant kinetic mechanism. Moreover, the values of about 0.17 signified Fickian diffusion but in a polydispersion spherical delivery system. From the experimental results, the values of $k_1$ with the ultrasound-treated ($k_1$ pH 1.2 = 20.10, $k_1$ pH 6.8 = 49.89) hydrogel beads were greater than those of the non-UTS-treated beads ($k_1$ pH 1.2 = 12.56, $k_1$ pH 6.8 = 47.26). This indicated that the

ultrasound-assisted release of the bioactive compounds occurred at a higher rate than the without-ultrasound release.

**Table 4.** Release kinetics of citronella oil bioactive compounds.

| System | Higuchi | | Ritger–Peppas Model | | | Peppas–Sahlin Model | | | | |
|---|---|---|---|---|---|---|---|---|---|---|
| | $k_h$ | $R^2$ | $k_1$ | n | $R^2$ | $k_1$ | $k_2$ | m | R/F | $R^2$ |
| pH 1.2 | | | | | | | | | | |
| Ultrasound | 11.40 | 0.948 | 20.10 | 0.267 | 0.993 | 24.69 | 0.21 | 0.11 | 0.0085 | 0.999 |
| Non-Ultrasound | 9.33 | 0.982 | 12.56 | 0.380 | 0.991 | 16.85 | 0.16 | 0.16 | 0.0107 | 0.999 |
| pH 6.8 | | | | | | | | | | |
| Ultrasound | 22.29 | 0.888 | 49.89 | 0.175 | 0.999 | 50.21 | 0.01 | 0.17 | 0.0003 | 0.999 |
| Non-Ultrasound | 21.64 | 0.890 | 47.26 | 0.176 | 0.999 | 50.42 | 0.07 | 0.14 | 0.0022 | 0.999 |

The results of the data adjustment performed using the Peppas–Sahlin equation showed a small relaxation factor value, but the Fickian diffusion had a large value in all the hydrogel bead systems. The value of the reaction rate constant ($k_1$), which was larger than the relaxation kinetic constant ($k_2$) from the Peppas–Sahlin equation as well as being positive indicated the dominant effect of Fickian diffusion and the small effect of the relaxation process on the release of the bioactive compounds in the citronella oil at pH 6.8. The R/F ratio indicated the contribution of diffusion and relaxation to the drug release. When R/F = 1, it creates a contribution that contributes to both erosion (relaxation) and diffusion. If R/F > 1, relaxation (erosion) dominates, while for R/F < 1, diffusion dominates. The R/F values of the UTS-treated (R/F pH1.2 = 0.0085, R/F pH6.8 = 0.0003) beads were smaller than those of the non-UTS-treated (R/F pH1.2 = 0.0107, R/F pH6.8 = 0.0022) beads, which means that UTS-treated beads had a higher F value than the non-UTS-treated beads. This affected the release of the bioactive compounds from the beads using UTS, as this resulted in more dominant diffusion effects and better stability compared to without UTS. Citronella oil was more easily released from the degraded Alg/Carr due to the breaking of the polymer compound chain, which was indicated by the smaller R/F value obtained using UTS. The solubility of the Alg/Carr increased when the ultrasonic waves broke the polymer chains. Therefore, the degraded polymer showed a lower ability to protect the encapsulated bioactive compounds and released them more rapidly.

*3.6. Proposed Mechanism of Ultrasound-Assisted Encapsulation Process*

Based on the explanation in the previous section, an ultrasound-assisted encapsulation mechanism was developed. Figure 7 shows the proposed mechanism in this study. As reported in several previous studies, the working principle of the ultrasonication process is that it causes an acoustic cavitation effect [26,47].

High-energy chemical reactions can be formed briefly in the presence of ultrasound waves through acoustic cavitation, which provides an interaction between the material and energy. The chemical effect of the ultrasonication process causes the molecules to interact so that chemical changes can occur. The mechanism of ultrasound-assisted polymer degradation occurs in two stages, namely homolytic bond cleavage and radical formation. The application of ultrasonic waves to the solution causes the molecules of the solution to oscillate relative to their position. When the ultrasonic wave energy provided is large enough, the wave strain can break the molecular bonds between solutions, and the gases dissolved in the solution will be trapped due to the solution molecules, whose bonds are broken when the density returns. This results in the formation of hollow balls or bubbles filled with trapped gas, which is known as the acoustic cavitation effect [27,48]. The bubbles will get bigger until they reach the size of the resonance bubbles, at which point the bubbles will collapse. The collapse of the cavitation bubble results in extreme conditions of very high temperature and pressure (up to 5000 °C and $5 \times 10^7$ Pa), which result in the breaking of polymer bonds. In polysaccharides, glycosidic bonds are easily broken due to this effect. In the second mechanism, free radicals ($H^+$ and $OH^-$) are formed due to the decomposition

of water during sonication. The concentrated energy released during bubble collapse creates extremely high local temperatures and pressures, such that the water molecules form free radicals in the gas phase [49].

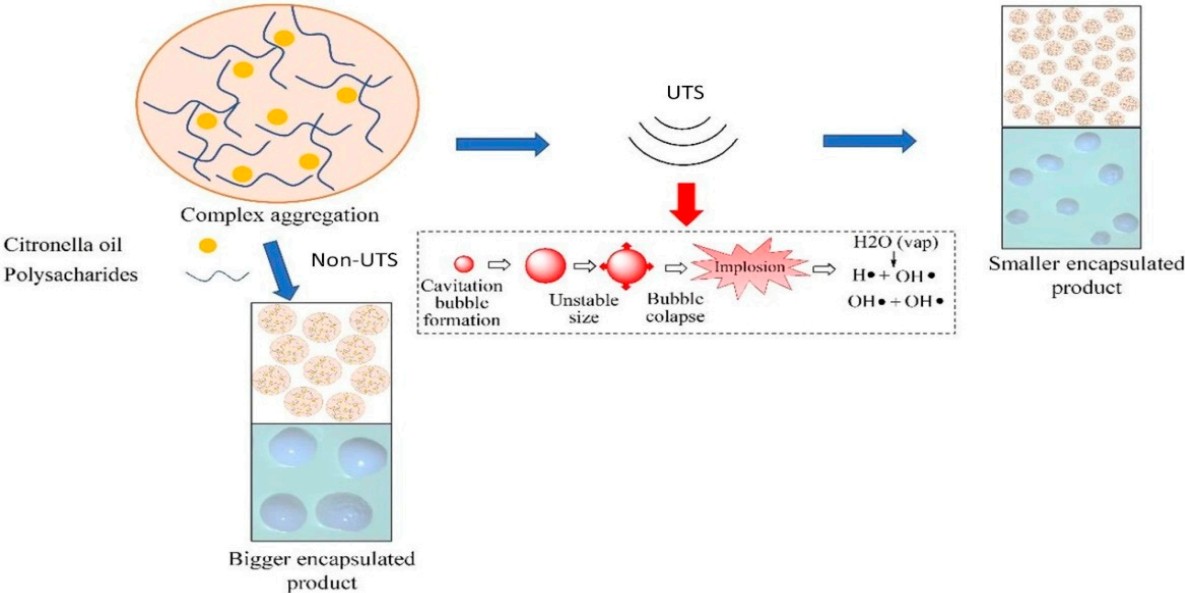

**Figure 7.** Encapsulation mechanism of citronella oil in Alg/Carr in the presence of ultrasonic irradiation.

$$H_2O \leftrightarrow H^+ + OH^- \tag{6}$$

$$OH^- + OH^- \leftrightarrow H_2O \tag{7}$$

$$H^+ + O_2 \leftrightarrow HO_2^- \tag{8}$$

$$H^+ + HO_2 \leftrightarrow H_2O_2 \tag{9}$$

$$HO_2^- + HO_2^- \leftrightarrow H_2O_2 + O_2 \tag{10}$$

$$H_2O + OH^- \leftrightarrow H_2O_2 + H^+ \tag{11}$$

Hydrogen peroxide ($H_2O_2$) and $OH^-$ ions are strong oxidizing agents that can degrade alginate polymers [49]. Ultrasound-induced depolymerization can reduce particle sizes because the resulting polymer is broken down by strong shear forces due to bubble collapse [27]. In depolymerization reactions, free radicals are formed in the fragment ends, and the shear stresses not only dissociate the bonds in the main chain but also separate the radicals formed, thus preventing their recombination. Cavitation and free radicals are very suitable for use in forming alginate and carrageenan polymers in the encapsulation of citronella oil. The carboxyl group in alginate and the sulfate group in carrageenan are easily reached by ultrasound and become a basis to form hydrogen bonds [15]. The strength of this linkage creates stability in the polysaccharide and in the microcapsules as a whole [22]. Strong hydrogen bonds formed due to ultrasonication-assisted cross-linking can provide better particle stability [50].

## 4. Conclusions

It was suspected that ultrasonication could increase the encapsulation efficiency of citronella oil in an Alg/Carr matrix to protect embedded bioactive components. In this research, ultrasound was successfully used to prepare Alg/Carr complex particles for the encapsulation of citronella oil. The FTIR spectra showed that the ultrasonication had a slight effect on the encapsulation of the citronella oil according to this study. The ultrasonication process caused the partial degradation of the biopolymers, which led to the formation of nanopores. The SEM images also revealed that the ultrasonication-treated

microcapsules were more porous, and the average microcapsule size was smaller and more uniform than that of the non-UTS treatment. On the other hand, the microcapsules had an irregular structure with the addition of SDS surfactant. However, the encapsulation efficiency of the citronella oil in the Alg/Carr was increased up to 95–97% with the presence of SDS and UTS treatment. The beads treated with ultrasound showed higher $k_1$ values using the Ritger–Peppas model and smaller R/F ratio using the Peppas–Sahlin model, indicating a diffusion-dominant mechanism of the release of the bioactive compounds, and the UTS-treated beads had better stability than the non-UTS-treated beads. The citronella oil was more easily released from the degraded Alg/Carr due to the breaking of the polymer compound chains, which was indicated by the smaller R/F value obtained using ultrasound. This research showed the great potential of ultrasonication treatment in controlling the release behavior of bioactive compounds from encapsulants in an effective and facile way.

**Author Contributions:** Conceptualization, A.P.; Methodology, A.P. and B.S.W.; Validation, A.P., B.S.W., A.H., A.D.A., S.F.C.M., N.R., D.P.U. and M.D.; Formal analysis, A.P. and B.S.W.; Experimental work, B.S.W.; Investigation, A.P., B.S.W., A.H., A.D.A., S.F.C.M., N.R., D.P.U. and M.D.; Resources, A.P.; Data curation, A.P. and B.S.W.; Writing—original draft preparation, A.P., B.S.W., A.H., A.D.A. and D.P.U.; Writing—review and editing, A.P., B.S.W., A.H., A.D.A. and D.P.U.; Visualization, A.P., B.S.W., A.H., A.D.A., S.F.C.M., N.R., D.P.U. and M.D.; Supervision, A.P.; Project administration, A.P.; Funding acquisition, A.P. All authors have read and agreed to the published version of the manuscript.

**Funding:** This research was funded by the Ministry of Education and Culture of the Republic of Indonesia through the Penelitian Dasar Unggulan Perguruan Tinggi (PDUPT) research scheme, grant number 394-04/UN7.6.1/PP/2023. The APC was funded by the Ministry of Education and Culture of the Republic of Indonesia.

**Acknowledgments:** The authors greatly acknowledge the financial support of the Ministry of Education and Culture of the Republic of Indonesia through the Penelitian Dasar Unggulan Perguruan Tinggi (PDUPT) research scheme.

**Conflicts of Interest:** The authors declare no conflict of interest.

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
