# Peer review of "Ultrasound-Assisted Encapsulation of Citronella Oil in Alginate/Carrageenan Beads: Characterization and Kinetic Models"

_2305-7084, doi:10.3390/chemengineering7010010_

Round 1

Reviewer 1 Report

The paper is discussing encapsulation of citronella oil using alginate/carrageenan (Alg/Carr). It is well-structured and contributes to the field. Some improvements could be made to polish the manuscript: 

Kindly change US abbreviation as it confused with the US (the country)

Authors should try to cite more articles in 2022 to update their literature

It is suggested to provide FTIR table reference for the selected wavenumber and their corresponding vibration to support the discussion

Author Response

Dear Distinguished Reviewer

Thank you for thorough and detailed review, we have completely answered your comments and suggestions in the revised manuscript. here our response to reviewer's comment. Please see the attachment.

Point 1: Kindly change US abbreviation as it confused with the US (the country)

 Response 1: Agreed, thank you for the suggestion. The abbreviation for ultrasound have been replaced with UTS instead of US in the revised manuscript.

Point 2: Authors should try to cite more articles in 2022 to update their literature

Response 2: Thank you for the recommendation. We have updated the references from articles published in 2022. Regarding the coverage of citronella encapsulation and ultrasound-assisted encapsulation is very limited in 2022, and we tried to extend in last 5 years. Here the list of updated references:

  1. Sari, I., Misrahanum, M., Faradilla, M., Ayuningsih, C. M., & Maysarah, H. (2022). Antibacterial Activity of Citronella Essential Oil from Cymbopogon nardus (L.) Rendle) Against Methicillin-Resistant Staphylococcus aureus. Indonesian Journal of Pharmaceutical and Clinical Research, 5(1), 16-22. (replace reference [1])
  2. Duarte, P. F., Wlodarkievicz, M. E., Nascimento, L. H., Puton, B. M. S., Fischer, B., Fernandes, I. A., ... & Junges, A. (2023). Microencapsulation of citronella essential oil (cymbopogon winterianus) with different wall materials using spray drying. Letters in Applied NanoBioScience, 12(3), 71. (replace reference [3])
  3. Trindade, L. A., Cordeiro, L. V., de Figuerêdo Silva, D., Figueiredo, P. T. R., de Pontes, M. L. C., de Oliveira Lima, E., & de Albuquerque Tavares Carvalho, A. (2022). The antifungal and antibiofilm activity of Cymbopogon nardus essential oil and citronellal on clinical strains of Candida albicans. Brazilian Journal of Microbiology, 1-10. (replace reference [4])
  4. Milinković Budinčić, J., Petrović, L., Đekić, L., Aleksić, M., Fraj, J., Popović, S., ... & Malenović, A. (2022). Chitosan/Sodium Dodecyl Sulfate Complexes for Microencapsulation of Vitamin E and Its Release Profile Understanding the Effect of Anionic Surfactant. Pharmaceuticals, 15(1), 54. (replace reference [20])

Point 3: It is suggested to provide FTIR table reference for the selected wavenumber and their corresponding vibration to support the discussion

Response 3: Agreed, the FTIR Table reference for the selected wavenumber has been provided in the revised manuscript in Table 2 Page 7.

Reviewer 2 Report

After reviewing the following manuscript, I consider that the work can be published after addressing the following points by the authors.

1.Introduction, the authors should improve the physicochemical description of the use of alginate and with which other polymers have been used to form micro and nano polymeric particles. 

2. The authors mention the use of alginate with carrageenan, but they must justify from their chemical affinity why they couple these two biopolymers. What would happen if alginate/chitosan is used to encapsulate this active ingredient?

3. In the experimental part, is it possible to know the molecular weight of carrageenan and alginate?,

4. The authors must explain the role that electrostatic interactions have in the process of formation of microparticles, their charging and release process. 

5. Is the surfactant part of the structure of the microparticle loaded with citronella?

6. The authors should explain how was the strategy to define the alginate and carrageenan ratio?,

Author Response

Dear Distinguished Reviewer

Thank you for thorough and detailed review, we have completely answered your comments and suggestions in the revised manuscript. here our response to reviewer's comment. Please see the attachment

Point 1: Introduction, the authors should improve the physicochemical description of the use of alginate and with which other polymers have been used to form micro and nano polymeric particles.

Response 1: Agreed, the introduction has been improved by providing the physicochemical description of alginate and their combination with other polymers to form micro polymers. This statement can be seen in Page 2 in Refference [14] Ramdhan et al., 2020

“Biopolymers that are often used in encapsulation are alginate and chitosan. In the for-mation of micro polymers, alginate-based formulations are used as texture modifier, stability enhancer and long-term efficacy of active compound enhancer due its biostability, biocompatible, biodegradable and non toxic delivery system. However, the mixture of al-ginate and chitosan is not soluble at pH 7 and well at pH 5”

Point 2: The authors mention the use of alginate with carrageenan, but they must justify from their chemical affinity why they couple these two biopolymers. What would happen if alginate/chitosan is used to encapsulate this active ingredient?

Response 2: Thank you for the recommendation, the description of chemical affinity of alginate and carragenan has been provided in revised manuscript in Page 2 paragraph 2 :

“Yu et al [15] reported that the formation of alginate/carrageenan hydrogels in CaCl2 produces a network structure with high affinity. besides that, in the presence of electrostatic attraction, alginate and Ca2+ form a bond network. At the same time, adjacent Ca2+ and carrageenan (OSO3-) form crosslinking networks between macromolecular crosslinking networks”

Point 3: In the experimental part, is it possible to know the molecular weight of carrageenan and alginate?

Response 3: Yes, it is possible to estimate the molecular weight of carrageenan and alginate, the reliable method to determine the molecular weight of biopolymer is Gel Permeation Chromatography (GPC). In this work, the alginate and carrageenan were obtained from reputable suppliers with known specification. We have mentioned the molecular weight of alginate and carrageenan used in this study acccording to their certificate of analysis in Page 3

“The chemical such as sodium alginate (molar mass 216.12 g/mol, molar weight 90-180 kDa with CAS Number 9005-38-3SIGMA-Aldrich, USA), carrageenan (molar weight 193-324 kDa),

Point 4: The authors must explain the role that electrostatic interactions have in the process of formation of microparticles, their charging and release process.

Response 4: the explanation of electrostatic interaction role in the formation of microparticles has been provided in the revised manuscript. First we mentioned in the introduction section page 2:

“Yu et al [15] reported that the formation of alginate/carrageenan hydrogels into CaCl2 produces a network structure with high affinity. besides that, in the presence of electrostatic attraction, alginate and Ca2+ form a bond network. At the same time, adjacent Ca2+ and carrageenan (OSO3-) form crosslinking networks between macromolecular crosslinking networks”.

And the explanation in the discussion section Page 9:

 The electrostatic charge of the polymer also plays important role in the encapsulate for-mation and citronellal oil charging and release process. Alginate and carrageenan have strong negative charge due to the presence of free carboxyl and hydroxyl as well as sulfated galactan in carrageenan. The negative sites of Alg/Carr interact with Ca2+ to form polymer network in the formation of encapsulate. The charging of citronellal oil into biopolymer encapsulation is assisted by the presence of surfactant due to its amphiphilic properties. The hydrophilic moiety of surfactant interacts with ionic sites of Alg/Carr otherwise the hydrophobic part of surfactant attracts the citronellal oil molecule. These hydro-philic/hydrophobic properties also controlled the release of citronellal oil from the Alg/Carr matrix [44]. Furthermore, the efficiency of encapsulation can be influenced by the irradia-tion of UTS.

  1. Stebbins, N. D., Faig, J. J., Yu, W., Guliyev, R., & Uhrich, K. E. (2015). Polyactives: controlled and sustained bioactive release via hydrolytic degradation. Biomaterials science, 3(8), 1171-1187.

Point 5: is the surfactant part of the structure of the microparticle loaded with citronella?.

Response 5: Yes, Surfactants (in this study is SDS) are used to modify carrageenan alginate which has hydrophilic properties to have hydrophobic part, so that the Alg/Carr matrix polymer can be mixed with hydrophobic citronellal oils. This explanation has been already rovided in the discrussion section of revised manuscript:

The charging of citronellal oil into biopolymer encapsulation is assisted by the presence of surfactant due to its amphiphilic properties. The hydrophilic moiety of surfactant interacts with ionic sites of Alg/Carr otherwise the hydrophobic part of surfactant attracts the citronellal oil molecule. These hydro-philic/hydrophobic properties also controlled the release of citronellal oil from the Alg/Carr matrix [28]. 

Refference:

Masina, N., Choonara, Y. E., Kumar, P., du Toit, L. C., Govender, M., Indermun, S., & Pillay, V. (2017). A review of the chemical modification techniques of starch. Carbohydrate Polymers, 157, 1226–1236. https://doi.org/10.1016/j.carbpol.2016.09.094

Point 6: The authors should explain how was the strategy to define the alginate and carrageenan ratio?,

Response 5: the alginate and carrageenan ration was defined according to the consideration of physcochemical properties of alginate and carrageenan to form desired encapsulation properties. We have performed preliminary research to simulate the formulation of alginate-carrageenan ratio as well as previous report study. This explanation has been provided in Page 3 in Refference [12] Postolovic et al., 2022 of revised manuscript.

“In the first step, determine the optimal Alg/Carr ratio used in this study is 2:1 [12]”. The comparison of alginate/carrageenan in Postolovic's study stated that the best combination in alginate composition was more than carrageenan
